# Siamese Content Loss Networks for Highly Imbalanced Medical Image Segmentation

**Brandon Mac**[1]                                                      BMAC@RYERSON.CA
**Alan R. Moody**[2,3]                                      ALAN.MOODY@SUNNYBROOK.CA
**April Khademi**[1,4]                                              AKHADEMI@RYERSON.CA

[1] *Image Analysis in Medicine Lab (IAMLAB), Ryerson University*

[2] *Department of Medical Imaging, University of Toronto*

[3] *Department of Medical Imaging, Sunnybrook Health Sciences Centre*

[4] *Keenan Research Centre for Biomedical Science, St. Michael's Hospital*

## Abstract

Automatic segmentation of white matter hyperintensities (WMHs) in magnetic resonance imaging (MRI) remains highly sought after due to the potential to streamline and alleviate clinical workflows. WMHs are small relative to whole acquired volume, which leads to class imbalance issues, and instability during the training process of many deep learning based solutions. To address this, we propose a method which is robust to effects of class imbalance, through incorporating multi-scale information in the training process. Our method consists of training an encoder-decoder neural network utilizing a Siamese network as an auxiliary loss function. These Siamese networks take in pairs of image pairs, input images masked with ground truth labels, and input images masked with predictions, and computes multi-resolution feature vector representations and provides gradient feedback in the form of a L2 norm. We leverage transfer learning in our Siamese network, and present positive results without need to further train. It was found these methods are more robust for training segmentation neural networks and provide greater generalizability. Our method was cross-validated on multi-center data, yielding significant overall agreement with manual annotations.

**Keywords:** Semantic Segmentation, White Matter Hyperintensities, Siamese Networks, Medical Imaging, Magnetic Resonance Imaging, Label Imbalance, Transfer Learning.

## 1. Introduction

White matter hyperintensities (WMH) in magnetic resonance (MR) images of a presumed vascular origin are understood to manifest due to a combination of local macroscopic tissue structure erosion and increased water content due to inflammation (Bakshi et al., 2005). Quantitative analysis of WMH in MR imaging is typically conducted in order to diagnose, and evaluate effectiveness of treatments. Typically, analysis is conducted manually utilizing specific criteria (Polman et al., 2011), visual scales (Pantoni et al., 2002), or manual delineations (Egger et al., 2017). The most informative analysis are manual delineations, as they provide volumetric information of lesion load and spatial distribution. However, acquisition of manual delineations are laborious, and are subject to high inter- and intra-rater variability. Reported voxel-wise agreement (F1 score) between radiologists have been

reported to range from a low of 0.66 (Egger et al., 2017) to a high of 0.83 (Steenwijk et al., 2013).

Recent advances in semantic segmentation in medical imaging has been largely in part to advent of deep learning methodologies. Most notably, U-Net style encoder-decoder fully convolutional networks (FCNs) has seen significant adoption in research community (Ronneberger et al., 2015). This is exemplified, as the top 11 teams of the MICCAI 2017 Grand Challenge for automatic segmentation of WMH all used U-Net inspired architectures. Typically, these architectures are trained utilizing a loss function which explicitly compares predictions to ground truths. However, WMHs are inherently class imbalanced due to their small size relative to acquired image. This skewed distribution affects training as predictions tend towards majority class. In a systematic study on impact of class imbalance on convolutional neural networks concluded that performance degrades, and is not just a relationship of number of training samples (Buda et al., 2018). Milletari et al. address class imbalance in their implementation of V-Net, which extends the U-Net from 2D to 3D domain, by utilizing a modification of Dice similarity coefficient as a loss function (Milletari et al., 2016). Investigations by Fidon et al. highlight potential limitations, namely that the loss function does not take advantage of multi-scale information (Fidon et al., 2017). Sudre et al. present a rebalancing strategy to allow more robust dice loss function (Sudre et al., 2017). Li et al. utilize a post-processing method in which predictions in the first and last 10% of the brain volume along the axial plane were discarded (Li et al., 2018).

## 1.1. Contribution

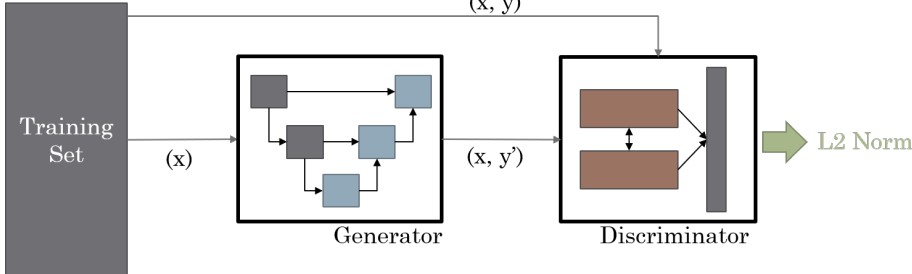

Figure 1: Overview of proposed training setup.

To overcome these challenges, this work proposes to utilize an auxiliary Siamese network to train a FCN segmentation model. Through this, multi-scale information is accounted for, which is not present in loss functions that explicitly compare predicted masks to ground truths. An U-Net style FCN with dense block convolutions was trained by only the gradients defined by an auxiliary Siamese network. Inspiration was drawn from the task of person re-identification (Re-ID), in which explicit comparison between images is not viable, and Siamese networks are typically used to encode information and measure similarity (Geng et al., 2016). Early layers of the VGG19 network (Simonyan and Zisserman, 2014) were used in our Siamese network, and feature mappings were sampled at different resolutions. For the loss function, we draw inspiration from style transfer implementations which utilize

content loss, in which feature mappings are flattened into column vectors and compared by their squared Euclidean distance (Gatys et al., 2015). The Siamese network measures the content similarity between pairs of masked input images; the input image multiplied by ground truth and input image multiplied by predicted mask. We developed this method on the dataset provided for MICCAI 2017 WMH Grand Challenge to have a standardized comparison to submitted entries. To verify generalization, we validate our results on manually segmented white matter lesions from the Canadian Atherosclerosis Imaging Network (CAIN). We compare our method by training the same FCN segmentation network, but trained using several loss for imbalanced segmentation tasks.

## 2. Method

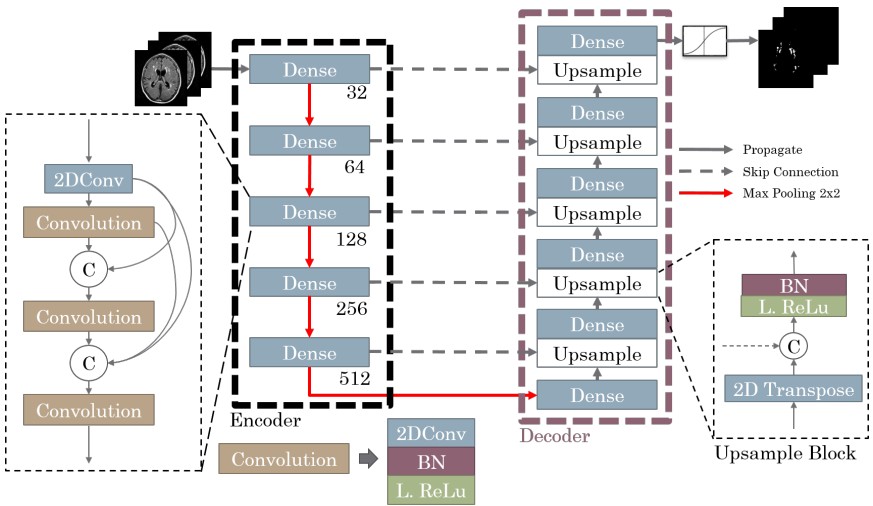

Figure 2: Generator Diagram.

### 2.1. Generator

The design of the segmentation model is focused on a fully convolutional encoder-decoder structure inspired by U-Net (Ronneberger et al., 2015). Formulation of convolution block consists of sequence of (2D) convolution, followed by batch normalization (Simonyan and Zisserman, 2014), and leaky ReLU activation (alpha = 0.1) (Xu et al., 2015). Choice of leaky ReLU over basic ReLU was to avoid "dying ReLU" problem in which some neurons become inactive and only output zero (Lu et al., 2019). All down sampling operations were conducted utilizing (2D) max pooling operations. Upsampling operation utilize (2d) transpose operations as shown in original U-Net (Ronneberger et al., 2015).

2D convolutions were utilized due to empirical results observed during the MICCAI 2017 WMH Grand Challenge, in which, submissions that utilized dilated or 3D convolutions placed near mid to low rankings (Kuijf et al., 2019). Choice of kernel size did not appear to affect performance, as such 3x3 kernels were utilized to reduce on number of parameters

to optimize. Dense blocks similar to ones in DenseNet were utilized, due to their properties to alleviate the vanishing-gradient problem, strengthen feature propagation, and encourage feature reuse (Huang et al., 2017). Initial dense block consisted of 32 filters, with the number of filters doubling with each max pooling operation. Number of dense block operations in the encoder portion was found by grid search and monitoring performance on a hold-out validation set, with a mirrored number of dense block plus one bottleneck dense block for the decoder, as shown in Figure 2.

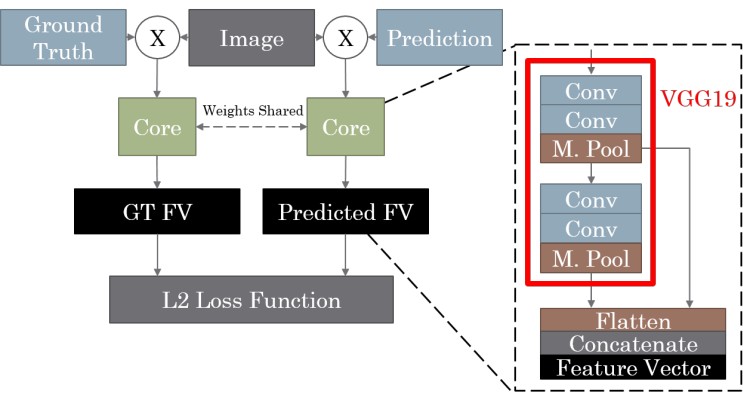

Figure 3: Discriminator Diagram.

## 2.2. Discriminator

The discriminator consists of a two path Siamese network. For every prediction mask generated by the segmentation network, the corresponding input image and ground truth mask is inputted into the discriminator. The masks are then multiplied to the input image, as shown in the two paths in Figure 3. Intention of multiplying the masks and input image together is to highlight the regions of interest and to allow network to jointly evaluate information from both images. We utilize weights from VGG19 network trained on ImageNet in our network (Simonyan and Zisserman, 2014). Choice to utilize VGG19 is due to the linear structure of the network progressively decreasing resolution of the convolution layer, thereby increasing receptive field. We sample the feature mappings at the max pooling layers of VGG19 network. We iteratively tested the number of feature mappings used and it was found that the first two max pooling layers yielded the best results, in which feature mapping dimensions were 112x112x64 and 56x56x128. Features in deeper layers are more domain specified towards natural images in ImageNet, and as such were unsuitable for this task. As well, the receptive field in deeper layers were also likely to be too large for this task, reflecting the small size of WMHs. In Equation (1), the function $f_c(.)$ is the transformation function representing the encoding of the masked images through the network and flattening to become feature vector representations. The feature vectors are then compared to each other by their squared Euclidean distance. Given as the following:

$$l_{mse}(f_c(x \cdot y), f_c(x \cdot y')) = \frac{1}{L} \sum_{i=1}^{L} \left( f_c(x \cdot y) - f_c(x \cdot y') \right)^2 \tag{1}$$

Where, x is the input image, y is the ground mask, y' is the predicted mask, and L is the length of the feature vector.

## 3. Experiment

### 3.1. Dataset

The training data used was provided by MICCAI as part of their 2017 grand challenge for WMH segmentation (Kuijf et al., 2019). The training set that was released publicly for development consisted of 60 MR image volumes from three institutes. Both T1 and fluid-attenuated inversion recovery (FLAIR) MR images were available, but in this implementation, only FLAIR images were used to correspond to acquired modality in cross-validation set. Of the 60 volumes, 48 volumes were randomly selected to be used to train, 6 volumes were used for validation of the aforementioned hyperparameters, and 6 volumes were used for cross-validation. We ensure even sampling between all centers when splitting data.

To cross-validate, 50 FLAIR volumes from Canadian Atherosclerosis Imaging Network (CAIN) were manually segmented by experienced raters using ITK-SNAP segmentation software (version 3.6.0) (Yushkevich and Gerig, 2017). All white matter lesions were manually delineated in a superior-to-inferior slice progression in the axial dimension. Once all white matter lesions were outlined on the axial plane, sagittal and coronal reconstructions were used to verify the segmentation and margins of the lesions.

### 3.2. Implementation Details

#### 3.2.1. Preprocessing

Whole volume was taken and normalized to an intensity between 0 and 1 by dividing by maximum intensity. The axial slices were then taken and resampled to 224 x 224 using default parameters of resize in python skikit-image library (Van der Walt et al., 2014). Images were then concatenated with itself twice in order to convert single channel intensity image into 3-channel RGB image. In order keep uniformity on all models, preprocessing steps outlined for VGG19 model were used (Simonyan and Zisserman, 2014). During training, random rotation augmentations up to a range of 30 degrees were applied to the image, utilizing generator functions in Keras.

#### 3.2.2. Training

For all models, 100 epochs were used during training with a batch size of 8 images and 100 batches per epoch. The ground truth was soft-binerized, meaning 0 values were set to 0.1 and 1 values were set to 0.9. Choice of this was to avoid exploding gradients when utilizing sigmoid as final output. Our initial learning rate was set to 0.001, and we utilize Adam optimizer with parameters set to $\beta_1 = 0.9$ and $\beta_2 = 0.999$ (Kingma and Ba, 2014). The models were developed on Keras-Tensorflow 2.0, and was trained in a single NVIDIA GTX 1080 Ti GPU.

### 3.2.3. Performance Metrics

Five metrics outlined by MICCAI 2017 WMH Grand Challenge for evaluation were used in order to have a standardized comparison to submitted entries (Kuijf et al., 2019). To measure performance in terms of class imbalance, we also define a normalized ratio which is the number of positive pixels in the ground truth, compared to the total image volume called the positive class density (PCD):

$$PCD = \frac{Sum(\#ofPositivePixels)}{Product(VolumeDimensions)}X100$$

This is to address the variation of the volume dimensions of acquired images. We utilize this metric as a normalized means to compare volume agreement between predicted masks and ground truths.

### 3.2.4. Benchmarks

We compare our proposed method to several loss functions commonly used in imbalanced semantic segmentation tasks. Dice loss as proposed by Milletari et al. makes modification the Dice score coefficient by introducing $\epsilon$ to allow for stability during training (Milletari et al., 2016). $\epsilon$ was set to 1 for benchmark comparison. Salehi et al. make modifications to this by proposing Tversky loss, in which $\alpha$ and $\beta$ terms are introduced as additional weightings for false positives and false negatives respectively (Salehi et al., 2017). In this investigation, $\alpha$ and $\beta$ were set to 0.3 and 0.7. To account for the bias of the Dice metric for larger volumes, Sudre et al. propose generalized Dice loss, in which the Dice loss is re-balanced by the squared volume of the ground truth (Sudre et al., 2017). For each of the aforementioned loss functions, we train the generator model with the same settings as mention prior in training section.

## 4. Results

In this section, we will describe the results of our proposed model to models trained with the mentioned benchmark loss functions. Visual inspection of segmentation masks, as shown in Figure 4, shows an overall greater sensitivity for lesion detection for proposed model. Models trained with Dice loss and Tversky loss appear to be under segmenting, as highlighted by the large number of false positives present in the first row of Figure 4. When validated on images derived from the same distribution as training set, the generalized Dice model and our proposed model appear to have similar performances on the MICCAI holdout. However, when evaluated with images outside the distribution of the training set, as shown in the last row of Figure 4, each of the benchmark models segment only a few of the lesions, missing the significant lesions on the bottom left. Whereas, our proposed model segments most of the lesions present. We observe this trend reflected in the average performances described in Table 1, namely that while the benchmark models have comparative performances on dataset derived from same distribution, when validated on a dataset outside the distribution, our proposed model shows significant improvement.

To observe the effects of class imbalance on the models, we assign bins to each volume in the CAIN independent validation set according to their positive class density and plot

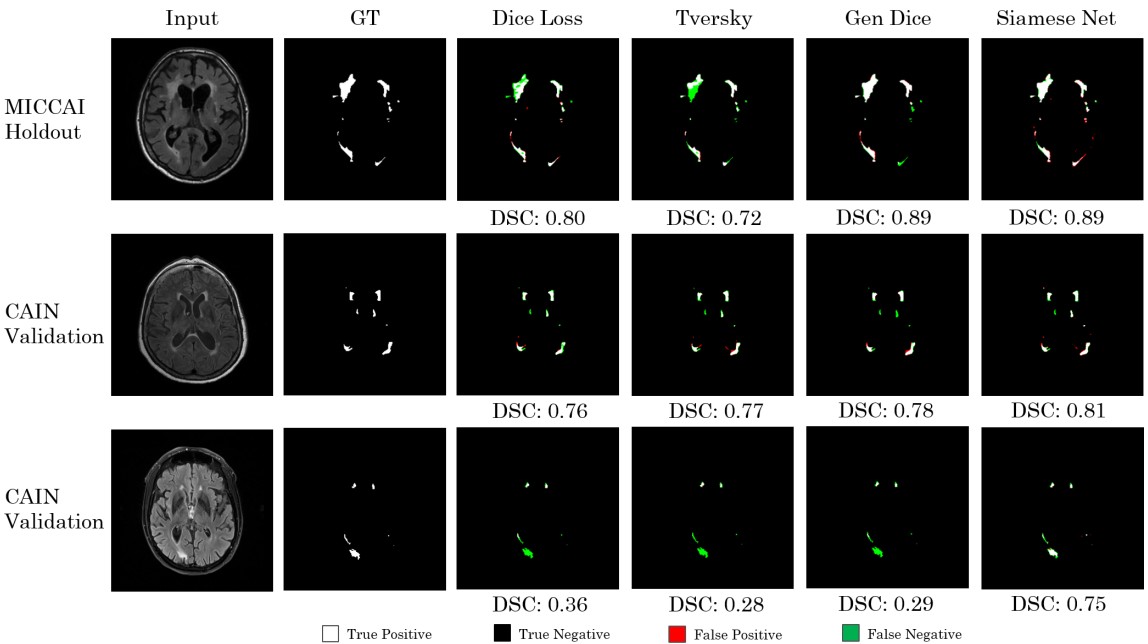

Figure 4: Prediction samples

Table 1: Summary of Performance Metrics

| Model | Dice ↑ | HD [2] ↓ | AVD [3] ↓ | L-Recall ↑ | L-F1 ↑ |
|---|---|---|---|---|---|
| Dataset: MICCAI Holdout Validation | | | | | |
| Dice Loss | 0.73(±0.07) | 9.33(±3.17) | 17.54(±10.77) | 0.46(±0.18) | 0.55(±0.18) |
| Tversky Loss | 0.78(±0.06) | 5.39(±2.28) | 10.79(±9.33) | 0.53(±0.13) | 0.63(±0.14) |
| Gen. Dice | 0.80(±0.06) | 4.43(±1.61) | 5.57(±5.33) | 0.61(±0.18) | 0.68(±0.17) |
| **Siamese Loss** | 0.79(±0.06) | 9.14(±9.96) | 11.82(±7.52) | 0.79(±0.14) | 0.62(±0.17) |
| *Leaderboard*[1] | *0.81* | *5.63* | *18.58* | *0.82* | *0.79* |
| Dataset: CAIN Independent Validation | | | | | |
| Dice Loss | 0.45(±0.22) | 26.76(±16.67) | 55.69(±38.24) | 0.48(±0.20) | 0.47(±0.16) |
| Tversky Loss | 0.43(±0.24) | 28.8(±20.54) | 52.52(±29.36) | 0.47(±0.25) | 0.48(±0.21) |
| Gen. Dice | 0.44(±0.27) | 21.40(±17.50) | 56.33(±28.79) | 0.50(±0.28) | 0.51(±0.25) |
| **Siamese Loss** | 0.52(±0.18) | 24.26(±16.40) | 39.76(±27.77) | 0.75(±0.15) | 0.54(±0.13) |

[1] MICCAI 2017 WMH Grand Challenge Leaderboard. Available at https://wmh.isi.uu.nl/results/
[2] HD refers to modifed Hausdorff distance (95th percentile) (mm)
[3] AVD refers to average volume difference (%)

versus the average volume difference between predicted volume and ground truth. Figure 5 depicts relatively low volume difference for our proposed model for most of bins. It is noted that two outliers exists in the 0.0 - 0.03 bin, in which the average volume difference is greater than 100%, indicating over-segmentation. For the benchmark models, there appears to be some volumes where the average volume difference is 100%, indicating no lesion was detected. Our proposed model on the other hand, detects at least some agreement to the ground truth.

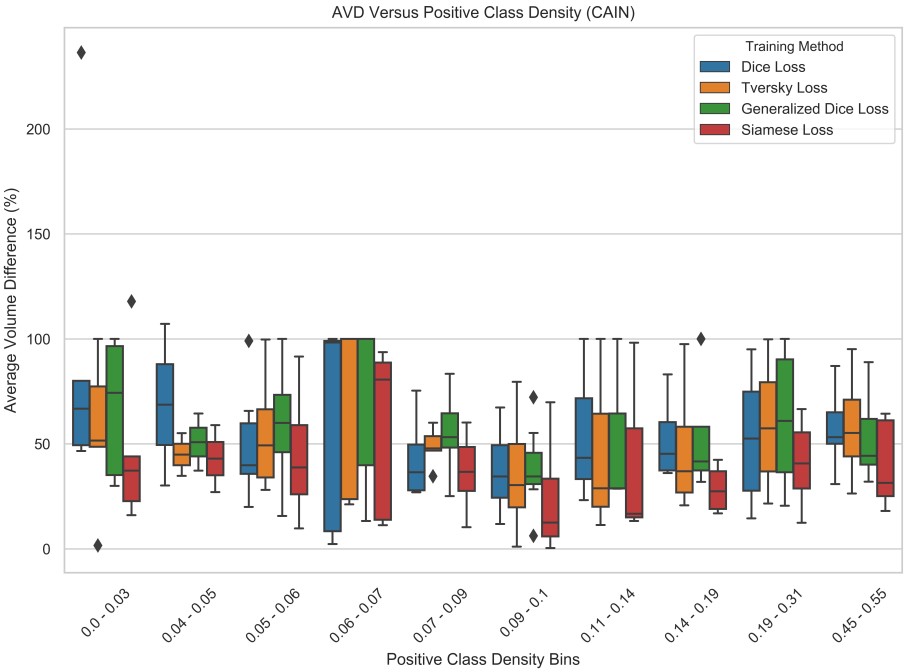

Figure 5: Average Volume Difference versus Positive Pixel Density

## 5. Discussion

In the previous section, we observe our proposed model generalizing well to data outside the distribution of the training set. We attribute this to the discriminator allowing for more contextual optimization of the generator weights. By evaluating the masked input images, texture features derived from the Siamese network allow for better comparison of information inherent in the pathology. The information taken at multiple depths of the pre-trained network represents the activation at multiple receptive fields, allowing for the latent vector to have a multi-scale representation. Whereas, overlap based optimization functions compare only masks, no additional information significant to the pathology is considered.

For this implementation, the core of the Siamese network is the VGG19 network pre-trained on ImageNet (Simonyan and Zisserman, 2014). Through tuning hyperparameters, as mentioned above, we found that the optimal layers to sample from were in the earlier layers, in which can be visually understood as edges and blobs (Zeiler and Fergus, 2014).

Fundamentally, ImageNet is a different domain from the medical images used in this implementation, namely ImageNet consists of natural colored images. We explore the aspect of fine tuning the VGG19 network to shift the domain more towards the present task as summarized in Table 2. The generator and discriminator were trained in a min-max fashion inspired by GAN type architectures (Goodfellow et al., 2014). Pre-trained weights of the Siamese network were unfrozen, and a small learning rate of 0.00001 was used for fine tuning. The objective of the generator was to minimize the L2 loss, while the discriminator sought to maximize it.

Table 2: Summary of Trained and Transfer Learned Siamese Networks

| Model | Dice ↑ | HD [1] ↓ | AVD [2] ↓ | L-Recall ↑ | L-F1 ↑ |
|---|---|---|---|---|---|
| Dataset: MICCAI Holdout Validation | | | | | |
| Trained SL | $0.51(\pm0.14)$ | $24.49(\pm14.07)$ | $32.32(\pm18.65)$ | $0.23(\pm0.17)$ | $0.24(\pm0.16)$ |
| **Siamese Loss** | $0.79(\pm0.06)$ | $9.14(\pm9.96)$ | $11.82(\pm7.52)$ | $0.79(\pm0.14)$ | $0.62(\pm0.17)$ |
| Dataset: CAIN Independent Validation | | | | | |
| Trained SL | $0.26(\pm0.21)$ | $61.12(\pm43.00)$ | $137.55(\pm384.34)$ | $0.16(\pm0.13)$ | $0.15(\pm0.13)$ |
| **Siamese Loss** | $0.52(\pm0.18)$ | $24.26(\pm16.40)$ | $39.76(\pm27.77)$ | $0.75(\pm0.15)$ | $0.54(\pm0.13)$ |

[1] HD refers to modifed Hausdorff distance (95th percentile)
[2] AVD refers to average volume difference (Percentage)

The result of fine tuning the weights yielded less than satisfactory results. Primarily, one reason could be due to the VGG19 network being highly parameterized, there is a lack of sufficient data samples to optimize the weights, despite the use of a low learning rate for fine tuning. Other reasons could be attributed to the GANs nature of the model. Since a min-max setup was used to optimize the weights, vanishing gradients could have affected the optimization of the model. A variety of other factors such as game setup and loss function selection could have attributed, however analysis of these design paradigms are beyond the scope of this paper and will be investigated in future works.

## 6. Conclusion

We present a training method which utilizes an auxiliary Siamese network to train a FCN segmentation model. Through this we found greater generalizability when compared to FCN model trained on loss functions which only evaluate only segmentation masks. By utilizing a Siamese network to evaluate the content loss between the masked images at each training step, we feedback multi-scale information in the training process. We found that this method allows for greater sensitivity, allowing for more robust evaluation across datasets and raters.

In this work, we leverage the use of transfer learning for our task, however we acknowledge the limitations in namely the domain specificity of the network and the over-parametrization contributing to computation overhead. We attempt to fine-tune the Siamese network through a min-max GANs setup, but found the results unsatisfactory. Future works

intends explore more of the design paradigms to allow for more efficient use of latent vector space and further refinement of features.

## Acknowledgments

We thank NSERC Discovery Grant Program for funding this research.

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
