# OpenReview forum: "Siamese Content Loss Networks for Highly Imbalanced Medical Image Segmentation"
_MIDL.io/2020/Conference — MIDL 2020_

### Official Review · AnonReviewer4 · 2020-03-13
**A nicely-proposed but although under-utilized implementation of  a novel loss function using content loss generated using auxiliary Siamese network**

**Rating:** 2
**Confidence:** 4
**Recommendation:** Oral, Poster

**Summary:**

This paepr presented an image segmentation framework in an generative model using a auxiliary siamese network trained discriminator as loss function, which produces improved segmentationa and more generalizability when compared to segmentation using the same generator but only with dice-based loss function.

- The main message from the paper is the proposal of a novel loss function using content loss, which is generated from a auxiliary Siamese network taking into account of the raw image, the ground truth segmentation, and the prediction.
- Transfer learning is used to initialize the segmentation network of the Siamese network with weights trained from ImageNet.
- Flattened feature map were used to calculate the loss for the discriminator

**Strengths:**

• The reason behind the decision for each points in the architecture design and parameters  are explained clearly, such as:
	• In the generator: choose 2D convolution over 3D convolution, the choice of kernel size (3x3), number of dense block operation in the encoder (using grid search),
	• choice of VGG19 in the discriminator
	• Soft-binarize ground truth to avoid gradient explode
	• The volume dimensions on top of the class-specific voxel number are taken into consideration when measuring class imbalance to compute the validation metric
• Many hyper-parameters are properly tuned, such as:
	• Number of dense-block operation used in the encoder
Number of feature mappings used at the max pooling layers

**Weaknesses:**

• Method comparison:
	• My biggest critics is for method comparison: only compared to raw dice coefficient? I would argue it would be more fair to compare the proposed loss function with other state-of-the-art loss function mentioned in the introduction, i.e. Milletari et al. 2016, Fidon et al. 2017, and Sudre et al. 2017.
	Especially, the proposed method seems still suffer from large class imbalance, although less severe then pure Dice loss (Figure 5-6)
	• Follow up on that, another of my critics is:  in page 8, result section, the author discussed that the "direct correlation between positive class density and segmentation performance" is "likely attributes to CNNs in general, as through kernel operations and max pooling, ﬁne spatial information is loss in the process." However, the Dice coefficient is known to be correlated to the volume size of the segmented mask. The "positive class density" in the axis is, by definition, correlated with the volume size of the segmented mask of the positive class/ Without correcting for that correlation, the dice shall naturally be higher for larger segmented volumes. Therefore, this conclusion might be misleading. Actually, this is indeed the reason that other methods which are cited by the author trying to adjust the raw dice before using as loss function.(The fact that in the baseline method, even though dice itself is used as the lost function, the evaluation using the same dice method showed unsatisfying results, indicating that the raw dice is not a good candidate for loss function)

• In Discriminator: a multiplication is used to feed the critics. The author stated that "*Intention of multiplying the masks and input image together is to highlight the regions of interest and to allow network to jointly evaluate information from both images.*" However, would this approach cause the critics to treat the over-segmentation and under-segmentation differently (e.g. over-segmentation get less penalty than under-segmentation, as less information is lost). Rather than of multiplication, how about instead do:
	• adding the mask as a new channel (similar to the idea of dense-block) to preserve full information from both the raw image and the mask
	• Or, using the mask as a weight to guide the critics to focus more on the masked region but less on the non-masked region (the weight itself can also be a learnable parameter)?
	• It seems the discriminator is not further trained in a generative adversarial (minmax) fashion, why not? Instead, pre-trained weights from ImageNet are directly used. This seriously limiting the feature maps that can be used, as well as the flexibility of input channels that can be used (e.g. using 3 consecutive slices as I proposed in the previous section).

• Preprocessing: 3-copy of input were concatenated to generate psudo-RGB images to utilize the transfer learning. That seems to be a bit under-utilization of the resources. Should it make more sense to instead extract 3 consecutive slices and concatenate them together in order to take advantage of the information in the adjacent neighbouring slices?


**Detailed Comments:**

• Generalizability: I appreciate that the author showed the independent validation on another dataset, to demonstrate the difficulty of generalizability of the CNN-based segmentation methods (Table 1). It would be great if the author could elaborate on that point a little bit more, and discuss about potential appraoches to alleviate that issue.
• In page 5 section 3.1 second paragraph, it might not be "cross-validation", but "independent-validation"
• In page 4 section 2.2: "to comparison to" => "to compare to"
• Table 1: since this table is mainly for method comparison between the proposed loss function and dice-loss I would suggest to reorganize the table rows: separate into three sections, each section have two rows: 1 using Dice loss, one use the proposed content loss
	• Also, it would be better to expand the Table 1 legend to mention the full name of the abbreviations in the table (e.g. HD, and AVD)
• Figure 6: The two subplots are showing similar information. If the authors do need to show both, I would suggest to combinethem together
Furthermore, it would be interesting to utilize the saved space of the second subplot to show some example images where the Dice loss totally failed (i.e. zero dice) while the proposed auxiliary siamase loss doesn't.

**Justification Of Rating:**

• The paper proposed a good combination of good practice into a newly proposed loss function
• The paper did a good job explaining  thoughts of the experimental design and implementation in great detailed
• The method comparison part of the paper need to be compared to other recently proposed state-of-the-art loss functions that the author cited in their introduction.
• The method evaluation in Figure 6 might be misleading.
The choice of combining the information in the mask (both ground truth and predicted) with the raw image information (which is simple multiplication) need to be adjusted and, if possible, can be improved

**Paper Type:**

both

**Questions To Address In The Rebuttal:**

• It would be more fair to compare the proposed loss function with other state-of-the-art loss function mentioned in the introduction, i.e. Milletari et al. 2016, Fidon et al. 2017, and Sudre et al. 2017.

• It would be great if the author could address the issue in Figure 6 about the effect of possitive class density to the dice similarity metric.

**Special Issue:**

yes

---

> ### Author Response · Authors · 2020-03-28
> **Thank you for your review.**
>
> We would like to thank the reviewer for their time and feedback.
>
> Addressing questions in review:
>
> During the rebuttal period, we have also conducted additional experiments that compared the proposed work to segmentation results for UNET alone based on different loss functions, including: the Dice loss, Tversky loss and Generalized Dice loss. To clarify, the Dice loss mentioned in the paper is the Dice loss proposed by Milletari et al. 2016, and we will clarify this in the revision. The generalized Wasserstein Dice score proposed by Fidon et al. introduces a weight matrix for their multi-class classification problem, but as the authors state for the binary case, the loss function reduces to a soft Dice loss. Instead of comparing to soft Dice loss, we opted to compare our results to Tversky loss proposed by Salehi et al. 2017 and Generalized Dice loss proposed by Sudre et al. 2017.
>
> We found similar trends as reported in Figure 6 when tested on the hold out set from the MICCAI data, with the generalized dice score providing the most competitive results to the proposed method.  However, when compared on the external clinical dataset, the proposed method continued to show the best generalization capabilities (similar to table 1).  We look forward to including these tables in figures in revised paper for this conference. See below some some new summary results we computed this week (to be included in revised paper):
>
> Performance on CAIN Independent-Validation (not used to train model)
> Tversky - DSC: 0.43
> Generalized Dice Loss - DSC: 0.44
> Siamese (proposed) - DSC: 0.52
>
> The original intention was to compare the effects of class imbalance on the performances of the models to the gold-standard metric used to evaluate segmentation. Although there is some correlation between the dice similarity metric and the size of the lesion mask, we are using this metric to *compare* between methods to see which is providing a more stable response overall lesion loads. If we were using this analysis to determine the performance of a single method, that it could be seen as biased. However, it is good from a comparison point of view.  Over the rebuttal period, we also investigated this analysis for some other metrics and for all the other UNET loss functions (see previous comments), including the average volume difference versus class density, which demonstrated that the proposed method is more stable (in terms of volume prediction) overall lesion loads.  We look forward to including this graph in the revision.  In future works, we will investigate other metrics as well.
>
> Addressing weaknesses in review:
>
> We thank the reviewer for these excellent ideas and suggestions, we are excited to incorporate them into our future works!!!  Many thanks! Masking the images was our initial attempt on trying to further refine segmentation masks from the generator based on the grayscale information, but you have suggested great ideas that will overcome some of the challenges that you have described.  Our initial thoughts were that the latent vectors of the discriminator would likely treat over-segmentation with more penalty than under-segmentation, hence we observe slight increase in false positives around the edges of our segmented lesions (see Figure 4) . But we agree the suggestions given would be perhaps better and we will look forward to experimenting with them in the near future.  Also, having the weights of the mask as a trainable parameter is an interesting prospect to be explored.
>
> An attempt was conducted to train the discriminator and generator in a min-max fashion to fine tune (learning rate for discriminator was set to 0.00001) the pre-trained weights, but found the results to be unsatisfactory, with performance significantly lower than any benchmark models. Comparing the trained discriminator to our proposed method, we found the average performances as the following: MICCAI - DSC: 0.51 vs 0.79 and CAIN - DSC: 0.26 vs 0.52. The results are likely attributed to various factors such as small data set, data homogeneity leading to mode collapse, and vanishing gradients due to game setup. The design paradigms associated were decided to exceed the scope of this paper and will be a topic of future investigations.
>
> It is an interesting prospect of encoding additional information from neighboring slices, and will be investigated further in future works. However, for this implementation we decided to utilize only 2D images because the acquisition of FLAIR MRI is often conducted in 2D with large slice thickness and then stacked to form a 3D image. [1] This is complemented by analysis by Kuijf et al. on results of the MICCAI 2017 WMH segmentation grand challenge, showing a significant decrease in performance when utilizing 2.5D or 3D operations.
>
> [1] R. Guerrero et al, "White matter hyperintensity and stroke lesion segmentation and differentiation using convolutional neural networks," NeuroImage. Clinical, vol. 17, pp. 918-934, 2018.

---

> > ### Comment · AnonReviewer4 · 2020-04-03
> > **I would like to thank the author for conducting the additional experiments during the rebuttal period to address my comments and questions**
> >
> > - It's good to get the clarification the Dice in VNet is used which is only calculated over the ROI region and discard the background pixels. Please ensure to clarify that in the final version.
> >
> > - We appreciate the additional experiments that the author reported to conduct during the rebuttal period. The additional comparison of loss function is also helpful to demonstrate the effective of the proposed method.
> >
> > - I agree that the method comparison is the main focus of the results in Figure 6. Please make sure to address appropriately in the final version that the difference of Dice in different positive class density might not be directly comparable (e.g. the trend shown in the line plot might especially cause such confusion)
> >
> > - The reporting the result of the additional experiments of the GAN might also be potentially useful, which might indicate the difficulty of training such min-max framework in a limited dataset.
> >
> > - The justifications of using single slices make sense.
> >
> > - It would be interesting to see result of future works incorporating the information from the mask in a better way. Thank you for considering these advices.
> >
> > Given the clarification given from the author during the rebuttal, I would be happy to accept the updated version of the paper, if the author can include these clarifications and incorporate the additional results from the additional experiment accompanied with corresponding discussions.

---

### Official Review · AnonReviewer2 · 2020-03-13
**overkill complex loss function instead of simple meaning of the computation**

**Rating:** 1
**Confidence:** 4

**Summary:**

The authors proposed a new architecture for the segmentation of white matter hyperintensities (WMHs) in MRI data. WMHs are small relative to whole acquired volume. This leads imbalanceness between training of CNN for the segmentation between WMHs and other parts. To overcome this imbalance problem and improve segmentation accuracy, the authors introduce a Siamese content loss to U-Net like architecture.

**Strengths:**

The new architecture consists of generator and discriminator. The generator, which is the U-Net like architecture, predict segmentation label. The discriminator, which is a Siamse Net, output L2 loss function as Siamese content loss. The two input to Siamse Net are images masked by ground truth and predicted label, respectively. From these input images, the Siamse Net compute feature vectors and evaluate the Euclidean distance between them as the loss. This loss is used for backpropagation in training of U-Net.

**Weaknesses:**

The methods do not technically sound. As the extraction of feature vectors of an input image in Siamse Net, the authors used the feature maps obtained from the first two max pooing layers in VGG Net, which is the part of  Siamse Net. It is known that the shallow layers extract only low-level features such that edges, blobs, and corners (ECCV paper,  https://arxiv.org/abs/1311.2901). Extracted feature vectors by the Siamse Net might represent these low-level geometrical feature on masked images. In addition to this property, regions of WMHs looks less texture in Figure 4. Therefore, the proposed loss function can be interpreted as a just edge- and counter-aware loss. This kind of loss function has been already reported as direct ways like below and other methods. The architecture is complex, but I think the computation can be more simple style.

Boundary loss for highly unbalanced segmentation:
 http://proceedings.mlr.press/v102/kervadec19a.html
Loss functions for image segmentation : https://github.com/JunMa11/SegLoss

Futhermore, for the evaluation of the proposed method, the authors trained U-Net with dice loss. Why they adopted the simple dice loss for imbalanced problem is unclear. For imbalanced problem, weighted dice loss is used in commonly. I think the selection of method for the comparison is unfair. Therefore, evaluation with this comparison does not make sense.


**Justification Of Rating:**

As I commented in the part of weakness, the meaning of the proposed loss function does not sound technically. Especially, justification of adopting Siamse Net is unclear. If there is convincing reason of this adopting, please describe it theoretically or experimentally. Moreover, the experimental evaluations look unfair validations.

**Paper Type:**

methodological development

**Special Issue:**

no

---

> ### Author Response · Authors · 2020-03-28
> **Thank you for your review.**
>
> We thank the reviewers for their comments and feedback.  The motivation for selecting the feature vectors from the siamese network is that we are able to incorporate multiresolution information from the lesions themselves.  While true that many of the filters may extract low level features such as edges, blobs and corners, there are 64 filters for the first layer and 128 filters for the second layer and therefore, much more than edge and corner information must be included.  Thanks to your great questions, during this rebuttal period, we did some exploratory analysis to investigate this further, by looking at the activation maps for several filters.  While some of the filters did indeed extract edge information, many extracted other things related to low frequency information, texture and other effects. The filter responses (activation maps) were not dominated by only the edges, and therefore, it is not the same as the provided reference above. By incorporating these additional features from the image, contextual information that is related to the pathology is incorporated into the optimization.
>
> The Siamese network allows the masked images to be encoded to a feature space such that comparing between ground truth and predicted masks allows for better measure of similarity based on the lesions fundamental low- and higher level components. Which in turn results in our model exhibiting greater generalizability when evaluated on data outside the training distribution . We agree that there areimprovements that can be made in terms of computation, and we will consider them in the future such as more efficient representation of latent vectors and/or refinement of features through GANs style optimization.
>
> During the rebuttal period, we have also conducted additional experiments that compared the proposed work to segmentation results for UNET alone based on different loss functions, including: including the Dice loss [2], Tversky loss [3] and Generalized Dice loss [1]. We found similar trends as reported in Figure 6 when tested on the hold out set from the MICCAI data, with the generalized dice score providing the most competitive results to the proposed method.  However, when compared on the external clinical dataset, the proposed method continued to show the best generalization capabilities (similar to table 1).  We look forward to including these tables in figures in revised paper for this conference.
>
> Siamese networks are commonly used for facial recognition, and its success in those applications was
> inspiration for this work.  Essentially, by comparing the raw feature activation maps, we are looking at the multiresolution information, for a variety of filters, which would allow us to describe pathology in a unique manner for both the segmentation and the ground truth lesions.  Essentially, we are decomposing lesions into their fundamental textural and low-level feature components, which should have some universality across datasets and lesions. This in turn is more representative of similarity between lesions rather than explicit comparisons based on segmentation outputs. We see the benefit of this as the model performs significantly better on the cross-validation set derived from images outside the distribution of the training set. Attributing to this is the use of feature vectors extracted from the masked images representing information inherent in the pathology.  We have in fact further experimentally validated this method, by comparing the proposed work to the UNET trained on loss functions mentioned above, and we have found comparable results on MICCAI holdout, but a significant improvement on the cross-validation set. Summarized in the following (and to be included in the revised paper):
>
> Performance on Cain Cross-Validation (not used to train model)
> Tversky - DSC: 0.43
> Generalized Dice Loss- DSC: 0.44
> Siamese - DSC: 0.52

---

### Official Review · AnonReviewer1 · 2020-03-13
**Metric learning loss for supervised highly imbalanced segmentation task.**

**Rating:** 2
**Confidence:** 5

**Summary:**

This paper presents a deep segmentation method of white matter hyperintensities (WMHs) in MRI. The architecture is based on a standard UNet model followed by a “Siamese” architecture that takes as input 1) the concatenated MR image and predicted segmentation mask and 2) the concatenated MR image and ground truth segmentation mask. The squared Euclidean distance between features extracted from the paired inputs of this “Siamese” network is used as a loss term to update the weights of the UNet architecture. The authors compare performance of their architecture to that achieved with a standard UNet trained on Dice loss. Evaluation is performed on FLAIR images of the MICCAI 2017 Grand Challenge and from the Canadian atherosclerosis imaging network (CAIN). This shows that the proposed architecture outperforms the standard UNet architecture.

**Strengths:**

-The general idea of improving segmentation tasks of small structures such as WMHs is of high interest.
-The paper is well written and the state of the art is clearly exposed.
-The main contribution of this paper consists in using a auxiliary loss term to train a UNet architecture is interesting. In this paper this loss term is derived from some distance between FLAIR images masked by the reference and predicted binary WMH maps.

**Weaknesses:**

-Description of this methodological contribution should be detailed. The authors should indeed explain how they backpropagate this ‘siamese’ loss term in the UNet architecture. The green arrow in Figure 1 is not explicit.

-The authors mention in the state of the art section that alternate loss terms than the DICE one have been proposed to better account for class imbalance issues (eg Sudre et al). It would be interesting to compare the proposed model with these architectures.

-The authors use a model replicating some part of the Siamese architecture, however, they do not try, if I understand well, to train this Siamese model, using correlated and uncorrelated pairs of images. Weights of the VGG19 core part of the Siamese model are those adjusted on ImageNet without any fine tuning on the MRI dataset.


**Justification Of Rating:**

The proposed ancillary loss is interesting, however, i am not convinced that the authors implemented the most efficient way (no training of the discriminator weights, separate training of the generator and discriminator part).  The paper lacks some methodological description to confirm the soundness of the proposed implementation. The authors also do not provide comparison with the results of the MICCAI 2017 grand challenge, this would add strength to the proposed method.

**Paper Type:**

both

**Questions To Address In The Rebuttal:**

-As suggested above, weights of the discriminator (Siamese) and generator (UNet) blocks could be trained on the MRI dataset, either separately  or simultaneously in a end-to-end fashion. Could the authors comment on this point?

-Could the authors clarify why they consider that their loss term allows feeding back “multi-scale” information in the training process? This indeed suggests, if I understand correctly, that the standard architecture of the UNet based on the Dice loss does not? Please clarify.


**Special Issue:**

no

---

> ### Author Response · Authors · 2020-03-28
> **Thank you for your review.**
>
> Thank you very much for your detailed comments and reviews. Please find our comments and responses below.
>
> “As suggested above, weights of the discriminator (Siamese) and generator (UNet) blocks could be trained on the MRI dataset, either separately  or simultaneously in a end-to-end fashion. Could the authors comment on this point?”
>
> The weights of the discriminator and generator can be trained separately in an adversarial fashion similar to GAN style architectures. By setting the objective function of the discriminator to maximize the L2 norm of the Siamese network, and setting the objective function of the generator to minimize the same objective, the weights can be optimized in a min-max fashion. The components of the networks would be trained sequentially, with training the discriminator first, then training the generator. Although, the reason for using VGG19 pre-trained on ImageNet was to address the small dataset issues, during the rebuttal period we have fine tuned the weights of the VGG19 network using the above setup with and setting a low learning rate of 0.00001 for the discriminator. What we found was a significantly lowered performance, with (MICCAI) - DSC = 0.51 and (CAIN) - DSC = 0.26 (compared to (MICCAI) - DSC = 0.79 and (CAIN) - DSC = 0.52 when no training was applied to discriminator). The poor results from training the weights of the discriminator can be attributed to small dataset problems which can lead to overfitting, data homogeneity leading to mode collapse, and vanishing gradients due to game setup. These design paradigms exceed the scope of this paper and are planned to be a subject of future works. We plan on including these results of training the discriminator as part of our analysis in the revised paper.
>
> “Could the authors clarify why they consider that their loss term allows feeding back “multi-scale” information in the training process? This indeed suggests, if I understand correctly, that the standard architecture of the UNet based on the Dice loss does not? Please clarify.”
>
> Our loss term consists of comparing the L2 difference between two latent vectors sampled from different layers in the VGG19 network. These latent vectors consist of the activation responses of different depths of the pre-trained network, which corresponds to multi-resolutional information since each layer describes different scales of information in the image (from low-level features such as edges, to higher level representations such as texture).
> At different depths, the receptive field becomes progressively larger due to max pooling, which allows the activation responses to describe multi-scale information efficiently, and it is this information that is used to fine tune the weights of the network through the L2 loss. This is in contrast to the dice loss, which merely compares the output segmentation to the ground truth (at the same native pixel resolution).  This will be clarified further in the revised version.
>
> Addressing weakness in review:
>
> “Description of this methodological contribution should be detailed. The authors should indeed explain how they backpropagate this ‘siamese’ loss term in the UNet architecture. The green arrow in Figure 1 is not explicit.”
>
> The objective function of the U-Net architecture is the minimization of the L2 norm of the vectors in the Siamese network. At each training step, the images are multiplied to the predicted and ground truth masks and each are fed into a separate path in the Siamese network. For each path of the Siamese network, the latent vector representation of the masked image is extracted and compared to each other. The mean squared error is calculated between the vectors and this loss is used to compute the gradients, which are then backpropagated to update weights of the generator network. Figure 1 will be changed to better reflect this in the revised version.

---

> > ### Comment · AnonReviewer1 · 2020-04-01
> > **Thanks for your reply**
> >
> > I would like to thank the authors for addressing my concerns.

---

### Official Review · AnonReviewer3 · 2020-03-15
**Siamese Content Loss Networks for Highly Imbalanced Medical Image Segmentation**

**Rating:** 3
**Confidence:** 5
**Recommendation:** Poster

**Summary:**

This method utilized a discriminator to calculate the segmentation loss values. The generator inputs the image and output a segmentation result. Then the image with ground truth and the image with the segmentation result will go through the discriminator which utilizes the perceptual loss to evaluate the segmentation results. The method is not novel, but it is interesting to do some experiments with this kind of setting.

**Strengths:**

1 The motivation for the proposed method is good. The design of this model is clear. It utilizes the GAN based idea to deal with the imbalanced problems.
2 The diagrams look good and help the reader to follow.



**Weaknesses:**

1. More comparisons with state-of-the-art methods should be added to demonstrate the effectiveness of the proposed method. The corresponding visualization results should be also added.
2. More loss functions should be explored. This author just utilized the L2 loss.


**Justification Of Rating:**

1 Methods are OK and the novelties are not significant.
2 The experiment part is not comprehensive. More comparison with state-of-the-art methods, including quantitatively and qualitatively should be added.

**Paper Type:**

methodological development

**Special Issue:**

no

---

> ### Author Response · Authors · 2020-03-28
> **Thank you for your review.**
>
> We would like to thank the reviewer for their time and valuable feedback. Please find our responses to your comments regarding weaknesses 1 and 2 below.
>
> In the submitted version, we compared the proposed method to the output of UNET with a dice loss, as shown in Figure 6. We focused on evaluation that shows that the proposed method: 1) is less sensitive to class imbalance issues than UNET alone with a dice loss (higher DSC over Positive Pixel Densities - Figure 6), and 2) has improved generalization capabilities through validation on an external, clinical dataset (CAIN) - DSC = 0.51 vs. DSC = 0.45 in Table 1. During the rebuttal period, we have also conducted additional experiments that compared the proposed work to segmentation results for UNET alone based on different loss functions, including: including the Dice loss [2], Tversky loss [3] and Generalized Dice loss [1]. We found similar trends as reported in Figure 6 when tested on the hold out set from the MICCAI data, with the generalized dice score providing the most competitive results to the proposed method.  However, when compared on the external clinical dataset, the proposed method continued to show the best generalization capabilities (similar to table 1).  We look forward to including these tables in figures in revised paper for this conference.
>
> In the near future, we will investigate other loss functions in the discriminator, along with other discriminator design paradigms - it was beyond the scope of this paper and are planned to be explored in future works. L2 loss is primarily chosen as it is commonly considered standard comparison between vectors, but we look forward to exploring this in a future journal paper.
>
> [1] C. H. Sudre et al, "Generalised Dice overlap as a deep learning loss function for highly unbalanced segmentations," 2017.
> [2] F. Milletari, N. Navab and S. Ahmadi, "V-net: Fully convolutional neural networks for volumetric medical image segmentation," in 2016, . DOI: 10.1109/3DV.2016.79.
> [3] S. S. M. Salehi, D. Erdogmus and A. Gholipour, "Tversky loss function for image segmentation using 3D fully convolutional deep networks," 2017.

---

### Meta-Review · Area_Chair1 · 2020-04-05
**MetaReview of Paper271 by AreaChair1**

**Rating:** 3
**Recommendation For Accepted Papers:** Poster

**Metareview:**

The paper presents an overall interesting development and work on the loss function to improve segmentation in the case of strong imbalance. Reviewers are consistently praising the quality of the rebuttal and the revised paper has the potential to be a much-improved version of interest to the MIDL community.

**Paper Type:**

methodological development

**Special Issue:**

no

---

### Decision · Program_Chairs · 2020-04-11

Accept